# Relationship between Circulating 25-Hydroxyvitamin D and Metabolic Syndrome in Chinese Adults: A Large Nationwide Longitudinal Study

**DOI:** 10.3390/nu16101480

**Published:** 2024-05-14

**Authors:** Mi Shu, Yue Xi, Jie Wu, Lai-Bao Zhuo, Yan Yan, Yi-Duo Yang, Yue-Yue Feng, Hua-Qiao Tan, Hui-Fang Yang, Yu-Ming Chen

**Affiliations:** 1Department of Epidemiology, School of Public Health, Sun Yat-sen University, Guangzhou 510080, China; shum7@mail2.sysu.edu.cn (M.S.); xiyue5@mail2.sysu.edu.cn (Y.X.); zhuolb@mail2.sysu.edu.cn (L.-B.Z.); yany36@mail2.sysu.edu.cn (Y.Y.); tanhq7@mail2.sysu.edu.cn (H.-Q.T.); 2Yibicom Health Management Center, Guangzhou 510530, China; wujie6370@cvte.com (J.W.); yangyiduo@cvte.com (Y.-D.Y.); fengyueyue@cvte.com (Y.-Y.F.)

**Keywords:** metabolic syndrome, serum 25-hydroxyvitamin D, Chinese adults, vitamin D deficiency, metabolic health

## Abstract

Objective: This study investigated the association of circulating levels of 25-hydroxyvitamin D (25[OH]D) with the risk of metabolic syndrome (MetS) and its components in adults. Methods: This nationwide cohort involved 23,810 Chinese adults attending annual health evaluations. Serum 25(OH)D levels, MetS status, and covariates were determined at each examination. Among them, 8146, 3310, and 1971 completed two, three, and more than three evaluations, respectively. A hybrid mixed-effects and Cox regression model was employed to determine the cross-sectional and longitudinal relationships. Results: The odds ratios (ORs) and 95% confidence intervals (CIs) of MetS were significantly lower in individuals within quartile 4 (vs. 1) of serum 25(OH)D for both between-individual (0.43 [0.35, 0.52]) and within-individual comparisons (0.60 [0.50, 0.73]), respectively (all *p*-trends < 0.001). Among the MetS components, the corresponding ORs (95% CI) in between- and within-individual comparisons were 0.40 (0.29, 0.54) and 0.26 (0.19, 0.36) for abdominal obesity, 0.49 (0.41, 0.58) and 0.78 (0.66, 0.93) for high triglycerides, 0.70 (0.59, 0.82) and 0.75 (0.64, 0.87) for hypertriglyceridemia, 0.48 (0.39, 0.59) and 0.87 (0.71, 1.07) for low HDL cholesterol, and 0.92 (0.76, 1.12) and 0.49 (0.41, 0.59) for hypertension, respectively. Decreased hazard ratios (95% CIs) in quartile 4 (vs. 1) of 25(OH)D were found for MetS (0.80 [0.65, 1.00]), high triglycerides (0.76 [0.62, 0.92]), abdominal obesity (0.77 [0.63, 0.96]), and low HDL cholesterol (0.64 [0.50, 0.81]). Conclusions: Decreased concentrations of serum 25(OH)D correlate significantly to a heightened MetS risk and specific components. Our findings underscore the potential preventive function of circulating vitamin D concerning metabolic disorders.

## 1. Introduction

Metabolic syndrome (MetS) is a medical condition distinguished by a collection of metabolic disorders, typically including abdominal obesity, dyslipidemia, hypertension, hyperglycemia, and insulin resistance [1]. Several studies have proposed potential mechanisms regarding the effect of vitamin D (VD) on MetS, such as stimulating renin gene expression [2,3], reducing insulin sensitivity [4], and influencing inflammatory responses [5]. The effect of VD on MetS may be related to the following factors: (1) In certain instances, a deficiency in VD is linked to endothelial dysfunction [6]. VD enhances nitric oxide levels, facilitating blood vessel dilation. VD deficiency can impact endothelial cells, leading to vascular dysfunction and disrupting vascular homeostasis [7]. (2) Reducing insulin resistance attributed to VD may involve phosphorylation of IRS1 at Ser307, enhanced expression of peroxisome proliferator-activated receptor gamma (PPAR-γ), and decreased phosphorylation of nuclear factor kappa B (NF-κB) p65 at Ser536 [8]. (3) Interaction between vitamin D and the VD receptor (VDR) reduces the levels of the proinflammatory cytokines, interleukin (IL)-2 and IL-12 [9]. (4) VD exerts a significant impact on lipid profiles through the suppressing the enzymatic activity of 3-Hydroxy-3-methylglutaryl-coenzyme A (HMG-CoA) reductase (HMGCR), which may enhance lipid metabolism in MetS [10,11]. Additionally, VD can serve as a fatty acid binding protein 4 (FABP4) inhibitor, offering therapeutic benefits for MetS by reducing bile secretion and regulating the PPAR signaling pathway [12].

The global prevalence of MetS is a significant health concern, affecting approximately 25% of the world’s population [13]. In the USA, this figure rises to 34.7% [14]. The situation in China is equally alarming, with a prevalence of 13.7% in 2001 [15] that surged to 33.9% in 2010 [14,16]. Particularly concerning is the annual growth rate of over 8% among individuals over 60 years [14]. This escalating trend, coupled with the high incidence rate, has significantly increased the health burden of MetS [13,17,18]. Therefore, the prevention and treatment of MetS is not just important but essential.

Several factors contribute to the development of MetS, including sex, age, ethnicity, and lifestyle changes, particularly unhealthy dietary habits leading to excessive energy intake and micronutrient deficiencies [19]. The relationship among micronutrients and VD has acquired significant attention due to its potential crucial position in metabolic diseases [20,21,22,23,24,25]. Negative associations have been discovered between VD and MetS. For instance, Lu et al. [26] reported a negative association through a cross-sectional investigation of 3289 participants aged 50–70. Similarly, Pham et al. [27] observed a beneficial association between serum 25(OH)D and MetS in a longitudinal analysis following 6682 participants for 7 years. Qi et al. [28] demonstrated in a meta-analysis that VD intervention reduced insulin resistance and hypertension but not lipids and HbA1c in adults with MetS. The results for the benefits of VD supplementation in MetS adults are inconclusive. However, this meta-analysis included randomized clinical trials (RCT) with petite sample sizes and most studies were conducted in Iran. Despite several studies investigating VD’s impact on MetS, the findings could be more consistent.

The investigation of the VD-MetS relationship has primarily relied on cross-sectional studies [21,22,23,24]. However, inconsistencies persist in the VD-MetS association [29]. Notably, a significant increase in MetS and VD deficiency among young and middle-aged adults has been observed. Yet, previous studies have predominantly focused on children [30,31], postmenopausal women [32,33], and older populations [24,34,35]. Therefore, our study’s focus on investigating the association of circulating 25(OH)D and MetS risk in young and middle-aged individuals is a novel and crucial contribution to the field.

Exploring the linkage between circulating 25(OH)D concentrations and the risks of MetS and its constituents in young and middle-aged Chinese individuals would help to provide new ideas for Mets prevention and management. The current study aims to understand the prevalence of MetS and VD deficiency in young and middle-aged Chinese people and explore the association of circulating 25(OH)D concentrations with the odds and incidence of MetS and its components. We hypothesized that people with decreased circulating 25(OH)D concentrations are more likely to be involved in MetS.

## 2. Study Participants and Methods

### 2.1. Study Participants

This cohort study encompassed participants who underwent annual health evalua-tions at the Yibicom Health Management Center in Guangzhou, China, covering five years from 2019 to 2024. Eligible participants are also required to be Chinese residents born in China, with an age of 18 or above. A total of 24,548 participants attended the health evaluations from 30 different cities across the country. Among them, we excluded those with missing data of MetS status or serum 25(OH)D (*n* = 560), 109 cases who did not meet the requirement of residence or the age range, and 69 cases with a malignant tumor. A total of 23,810 participants were included in the cross-sectional investigation. The longitudinal analyses contained 8146, 3310, and 1971 individuals who had completed two, three, and more than three annual health evaluations (Figure 1 and Figure 2).

The study protocol adhered to the ethical principles delineated in the 1975 Declaration of Helsinki and acquired permission from the Research Ethics Committee of Sun Yat-Sen University’s School of Public Health (No. 2024-026). 

### 2.2. Research Method

#### 2.2.1. Data Collection

Trained medical technicians collected data through a standardized electronic questionnaire and physical examination system during the physical examination. This questionnaire captured detailed information on demographics (age and gender), lifestyle factors (smoking and drinking history), and medical history (dyslipidemia, hypertension, coronary heart disease, stroke, and diabetes mellitus). Blood collection dates were automatically logged and categorized by season: spring, summer, autumn, and winter. Smoking status was defined as having smoked for six months or more, while alcohol consumption was considered as drinking daily for at least six consecutive months.

#### 2.2.2. Physical Examination

Physical measurements were performed using an Omron electronic scale (model HNH-219, Tokyo, Japan) and an Omron electronic sphygmomanometer (model HBP-9021, Tokyo, Japan), adhering to international standards. Participants were required to wear a medical gown, stand barefoot and upright, and remove their shoes for precise height and weight assessments to the nearest 0.1 cm and 0.1 kg. Their body mass index (BMI) was then obtained by dividing their body weight (kg) by height^2^ (m). We determined the blood pressure for the participants after a 5 min rest, with no prior intake of coffee, alcohol, smoking, or emotional disturbance. Two readings were taken on the right arm, 1–2 min apart, and their average was recorded. We measured their waist circumference (WC) at the center between the iliac crest and the lower rib margin. Each measurement was taken twice, averaged, and rounded to the closest 0.1 cm. 

#### 2.2.3. Laboratory Examination

Laboratory tests were performed at the Yibicom Health Management Center, accredited under ISO15189 [36] (registration number: CNASMT0691). Subjects provided fasting venous blood samples after an 8–12 h fast and these were processed within 2 h. Blood samples were collected using the Abbott i2000SR chemiluminescent immunoassay analyzer (Abbott, Green Oaks, IL, USA) for serum 25(OH)D concentrations with a coefficient of variation (CV) of 6.25%. The hexokinase method measured fasting blood glucose (FBG) levels on an Abbott C16000 automated biochemistry analyzer. Fasting insulin levels were collected via a chemiluminescent microparticle immunoassay (CMIA). The Homeostatic Model Assessment for Insulin Resistance (HOMA-IR), calculated as fasting insulin (μU/mL) × fasting glucose (mmol/L)/22.5, was used to evaluate insulin resistance. Lipid profiles, including total triglycerides (TG) and cholesterol (TC) and HDL and LDL cholesterol (HDL-c and LDL-c), were assessed utilizing enzymatic methods. At the same time, glycated hemoglobin was analyzed through the high-performance liquid chromatography (HPLC) method on an analyzer (Bio-Rad D-10, Schiltigheim, France). 

### 2.3. Relevant Definitions

MetS status was classified according to the National Cholesterol Education Program Adult Treatment Panel III (NCEP-ATP III) criteria adapted for Asian Chinese individuals. Diagnosis required over two of the following five conditions: (1) abdominal obesity, with a WC ≥ 90/80 cm for men/women; (2) high TG (≥1.7 mmol/L) or ongoing treatment for dyslipidemia; (3) low HDL-c (<1.0/1.3 mmol/L for men/women); (4) hypertension, defined as diastolic blood pressure ≥ 85 mmHg or systolic blood pressure ≥ 130 mmHg, or the utilization of antihypertensive medication; and (5) hyperglycemia, represented by FBG ≥ 5.6 mmol/L or ongoing antidiabetic medication usage [37]. Incident Mets or its components were defined as detecting the condition at either follow-up in those participants without the condition at baseline. We defined VD deficiency according to the cutoffs of the Institute of Medicine of the National Academy of Sciences (IOM) [38] and the China Health Industry Standard [39] (VD deficiency < 12 ng/mL, insufficiency: 12–20 ng/mL, and sufficiency: ≥20 ng/mL for serum 25[OH]D). The VD distribution among the study participants was categorized into age- and sex-adjusted quartiles, with the following breakpoints: Q1 group (11.9 [3.1] ng/mL, 6663 cases), Q2 group (16.8 [3.6] ng/mL, 6172 cases), Q3 group (20.8 [4.2] ng/mL, 5865 cases), and Q4 group (28.0 [6.6] ng/mL, 5110 cases).

### 2.4. Statistical Methods

For this study, baseline data from the initial health evaluation were utilized to outline the participants’ essential characteristics. Participants were categorized into sex- and age-stratified quartiles (Q1–Q4) based on baseline serum 25(OH)D levels, with Q1 serving as the reference group. To examine distinctions in characteristics among groups, we employed an analysis of variance and Chi-square test. Survival analysis was conducted to explore the risk of MetS in quartiles 2–4 (vs. 1) of average 25(OH)D values before MetS onset in participants without MetS at baseline, and hazard ratios (HRs) and 95% confidence intervals (CIs) were calculated by employing Cox regression in R studio (2022.07.1 Build 554) (R Studio, Boston, MA, USA). A logistic hybrid mix-effect model was employed to capture both the intra- and inter-subject relationships between VD and MetS [40]. Odds ratios (ORs) and 95% confidence intervals (CIs) were calculated to describe the estimated effect sizes for the associations between the quartiles of intra-subject (
$X_{ijt}-\bar{X}$
*_i_*) and inter-subject variations (
$\bar{X}$
*_i_*) of 25(OH)D with the status of MetS (MetS*_t_*) at various time points (*t*), using the “*xtreg*” program of Stata/SE 17.0 software (StataCorp LLC, College Station, TX, USA), which indicate cross-sectional and longitudinal relationships between VD and the presence of MetS (Equation (1)) [40]. *p*-trend values were calculated by treating the quartiles as a continuous variable. We defined the statistical significance at a *p*-value < 0.05 (two-sided).

(1)
$$Y_{it}=b_{0}+\overset{J}{\underset{j=1}{\sum }}b_{Bj}{\bar{X}}_{i}+\overset{J}{\underset{j=1}{\sum }}b_{Wj}\left(X_{ijt}-{\bar{X}}_{i}\right)+\ldots \ldots$$


In this model [40], *Y_it_* represents the dependent (outcome) variable for participant *i* at time point *t*; b_0_ denotes the intercept; *b_Wj_* is the regression coefficient indicating the within-individual component of the association; and *b_Bj_* signifies the regression coefficient capturing the between-individual component of the association. Furthermore, *X_ijt_* is defined as the independent (exposure) variable *j* for individual *i* at time point *t*, while 
$\bar{X}$
*_i_* represents the mean of the exposure variable *X* across all observed time points for each participant.

## 3. Results

### 3.1. Participant Characteristics

The investigation involved 23,810 participants, comprising 12,596 males and 11,214 females. The mean (SD) values of age and serum 25(OH)D were 43.6 (13.3) years and 18.8 (7.3) ng/mL. A total of 17.9% (*n* = 4260), 42.9% (*n* = 10,213), and 39.2% (*n* = 9337) of participants had VD deficiency, insufficiency, and sufficiency based on serum 25(OH)D, respectively. VD deficiency and insufficiency were more frequent in women than men (64.0% vs. 57.9%). The VD deficiency was actually lowest in summer and fall, and while it did decrease with age, it remained most prevalent in the 18–44 age group. (Appendix A).

Individuals in quartile 4 (vs. 1) of serum 25(OH)D had lower levels of diastolic blood pressure, pulse, BMI, WC, FBG, FBI, HOMA-IR, HbA1c, PBG, and TG (*p*-all < 0.05). The study also showed that MetS prevalence was significantly higher in quartile 1 (vs. 4) of serum 25(OH)D levels (30.1% vs. 23.8%) (Table 1).

### 3.2. Relationships between Serum 25(OH)D Levels and MetS Risk—Hybrid Mixed-Effect Model

The mixed-effects regression model revealed a significant beneficial association of inter-individual serum VD variations with the odds and risks of MetS and its components, including high WC, high TG, low HDL-C, and hyperglycemia. Specifically, participants in quartile 4 of serum 25(OH)D showed a notably lower likelihood of detecting MetS, high WC, high TG, low HDL-C, and hyperglycemia than those in quartile 1 (the lowest).

After adjusting for covariates, such as sex, age, season, smoking, and drinking habits, the results from the cross-sectional comparisons (between-subject relationship) showed that the highest (vs. lowest) quartile of VD exhibited a 57%, 60%, 51%, 52%, and 30% decrease in the likelihood of detecting MetS, high WC, high TG, low HDL-C, and hyperglycemia, respectively (all *p*-trends < 0.001). Longitudinal analyses (within-subject relationship) revealed that participants in quartile 4 (vs. 1) of VD had 40%, 74%, 22%, 51%, and 25% lower risks of developing MetS, high WC, HTG, hypertension, and hyperglycemia, respectively (all *p*-trends < 0.05). There is no significant between-subject relationship for hypertension and within-subject relationship for low HDL-c (Table 2).

### 3.3. Mean 25 Hydroxyvitamin D Values and the Incidence of MetS—Cox Regression Model

In quartile 4 (vs. 1) of 25(OH)D, the adjusted hazard ratios (95% CI) for the incidence of MetS, high WC, high TG, and low HDL-c are 0.80 (0.65, 1.00), 0.77 (0.63, 0.96), 0.76 (0.62, 0.92), and 0.64 (0.50, 0.81), respectively (all *p*-trends < 0.05). However, the beneficial association of 25(OH)D was not significant in relation to the risks of hypertension and hyperglycemia (Table 3).

## 4. Discussion

In this population-based study with a large sample size, we explored the associations between circulating 25(OH)D concentrations with both the odds and risk of MetS and its components among Chinese adults nationwide. Serum 25(OH)D deficiency and insufficiency were prevalent in China, with a proportion of 17.9% and 42.9%, respectively, and the MetS prevalence was 27.5%. Moreover, significant beneficial associations existed between circulating 25(OH)D concentrations and the odds and incidence of MetS and most of its components. These findings imply that VD deficiency and insufficiency may be essential in developing MetS among the Chinese population.

### 4.1. Prevalence of VD Deficiency and Insufficiency and VD Supplement Users in Chinese Adults

Many studies report prevalent VD deficiency and insufficiency in Chinese adults. Our findings showed 17.9% VD deficiency and 42.9% VD insufficiency among the middle-aged population. Notable variance occurred in the prevalence of VD deficiency and insufficiency among various Chinese populations, such as 28.6% deficiency and 37.6% insufficiency in urban middle-aged individuals aged 35–60 by Yin et al. [41], 78.3% deficiency (<20 ng/mL) in adolescents aged 14–28 [31], and 69.2% and 24.4% of deficiency and insufficiency among the elderly population aged 50–70 [26], respectively. This discrepancy may be due to the inconsistency in age and place of residence of the research participants.

VD insufficiency is prevalent in the Chinese population. However, domestic surveys on VD supplementation have focused on children [42] and pregnant women [43,44], with fewer surveys on young or middle-aged adults, with a proportion of VD supplement usage ranging between 4.8% to 50.9%. Our study did not collect data on supplement use, thereby limiting our ability to assess the impact of vitamin D supplements on serum levels and the prevalence of deficiency and insufficiency. Considering substantially high VD deficiency and insufficiency in this population, current VD usage is not enough to control this problem. Future studies should aim to fill this gap by examining the effects of supplementation across a broader demographic.

### 4.2. Serum 25(OH)D and Metabolic Syndrome

Our findings elucidate that individuals with serum 25(OH)D deficiency are more inclined to develop MetS in cross-sectional and longitudinal analyses. Importantly, this beneficial association persisted significantly even after the adjustments for sex, age, and season. This finding is in line with a comprehensive meta-analysis [45] that included 28 studies and 99,745 participants, indicating a significant reduction in the MetS prevalence with elevated serum 25(OH)D concentrations. Furthermore, large-scale, cross-sectional studies from the USA [46,47,48], the United Kingdom [49], Taiwan, and China [50] have reported similar findings among adults. However, there are differences in the results reported by Mansouri and Mehri in Iran [51,52]. These discrepancies may be because Iranian women typically cover their skin with clothing, limiting sunlight-induced VD synthesis. Intriguingly, our findings indicate that the beneficial association between VD and MetS is more pronounced in the ORs derived from hybrid logistic mixed models than in the HRs from Cox regression models. The ORs were calculated using hybrid logistic mixed-effect models to account for changes in VD and MetS status across all repeated measurements. Conversely, the Cox regression model defined incident cases based on the detection of MetS at any follow-up visit among initially MetS-free individuals, using average VD levels prior to the onset of MetS. This approach overlooked fluctuations in VD levels and subsequent changes in MetS status. Similar situations were also noted for some MeS components (e.g., hypertension and hyperglycemia). Therefore, the hybrid logistic mixed-effect models likely offer more robust evidence of the VD-MetS relationship than the Cox regression models.

### 4.3. Serum 25(OH)D and Abdominal Obesity and Dyslipidemia

Furthermore, our findings revealed a substantial beneficial correlation between serum 25(OH)D levels, abdominal obesity, and high TG. This finding is consistent with a prospective study on 3240 middle-aged and older adults [53]. In addition, Ford et al. [47] found a significant inverse correlation between 25(OH)D quintiles and the odds of abdominal obesity and hypertriglyceridemia in a cross-sectional analysis of 8421 20-year-old American males and nonpregnant females. Additionally, similar findings were reported in the studies conducted by Kayaniyil et al. [54] in Canada, which showed higher risks for type 2 diabetes, and Chacko et al. [33] in American postmenopausal women. Several possible factors may contribute to this correlation. Firstly, the resistance to insulin induced by free fatty acids could lead to the sequestration of VD within adipose tissue, which can be counteracted by serum 25(OH)D concentrations [55]. Alternatively, it suggests that the association may be related to obesity, often accompanied by unhealthy diet, sedentary lifestyles, reduced outdoor activity, and shorter sunlight exposure [56]. Perticone et al. [57] observed that patients on a very low-calorie ketogenic diet (VLCKD) experienced a significant increase in serum 25(OH)D levels from 18.4 (SD 5.9) to 29.3 (SD 6.8) ng/mL, attributed to a marked decrease in adiposity among the obese population. Vitamin D stored within the fat is released. The influence of vitamin D on lipid profiles involves a multifaceted mechanism [58], encompassing both the activation of its receptors and the reduction of lipid accumulation. When serum vitamin D concentrations surpass a specific limit, its beneficial impacts on lipid parameters become significantly more evident.

### 4.4. Serum 25(OH)D and Hyperglycemia

In the investigation of the association between VD and hyperglycemia, it was observed that FBG, FBI, HOMA-IR, HbA1c, and PBG levels were increased in participants with the highest quartile (Q4 vs. Q1) of circulating 25(OH)D. Both cross-sectional and longitudinal studies have documented a significant correlation between serum 25(OH)D and hyperglycemia. Cross-sectional studies in 3289 Chinese middle-aged individuals [26] and in 8421 American men and nonpregnant women [47] revealed a significant relationship between low 25(OH)D concentrations and increased insulin resistance risk. Cohort studies found lower concentrations of 25(OH)D were associated with increased blood glucose and insulin resistance in 524 non-diabetic subjects aged 40 to 69 years in the UK [59] and 4164 Australian adults [60]. Sciacqua et al. [61] also identified an inverse correlation between blood 25(OH)D levels and glucose tolerance. VD influences glucose and insulin homeostasis by regulating immune responses and β-cell function, enhancing peripheral insulin sensitivity, and ameliorating proinflammatory conditions [62]. Related studies suggest elevated PTH [35,61] and obesity [56] may also impact VD’s biological effects. The biological effects of VD are multifactorial and complex.

### 4.5. Serum 25(OH)D and Hypertension

Our longitudinal analysis found a fascinating negative correlation between serum VD and the risk of hypertension. Consistent findings were observed regarding the beneficial association between VD and hypertension [28,60,63,64]. A study on the removal of the VD receptor (VDR) gene in mice showed heightened activity in the renin–angiotensin–aldosterone system (RAAS) and elevated blood pressure levels [65]. A meta-analysis of randomized trials revealed that VD supplementation did not affect blood pressure in childhood populations [66]. A ten percent increase in serum 25(OH)D levels was not related to blood pressure (SBP) in 146,581 Europeans [67]. Therefore, large-sample, multicenter, randomized controlled trials are warranted to further understand the complex relationship between VD and blood pressure. 

### 4.6. Strengths and Weaknesses

To the authors’ knowledge, limited studies have evaluated the association between the distribution of circulating 25(OH)D concentrations and MetS risk in Chinese adults (≥18 years). Our study aims to contribute to understanding this relationship and providing approaches for preventing MetS and VD deficiency. Additionally, our study benefits from a large and diverse sample size, encompassing participants from different genders, age groups, seasons, and geographical regions across the nation. This enhances the representativeness of our findings and facilitates their generalizability. Notably, the repeated measures of serum 25(OH)D and MetS outcomes permitted our study to employ hybrid mixed-effects models to examine longitudinal outcomes, enabling us to observe dynamic relationships and individual variations between variables accurately. This approach enhances the reliability of inference. 

However, our study has some limitations. Firstly, because of the retrospective nature of the data analyses, we could not exclude the potential effects of lifestyle habits and nutritional supplements on VD levels, such as outdoor activities, dietary patterns (particularly the consumption of fatty fish or fortified foods), and VD supplementation due to unavailable data. Additionally, as an observational study, we cannot establish causality, although we minimized categorization bias in exposure indicators by assessing changes in VD levels at multiple time points.

## 5. Conclusions

Our study has elucidated that lower serum 25(OH)D concentrations are linked to raised risks of MetS and several of its components in young and middle-aged adults in China. The robust evidence provided by our longitudinal study design, which utilized a large nationwide sample, supports these findings. However, further well-designed randomized controlled trials are necessary to establish a causal relationship. Additionally, greater focus on the adverse effects of VD deficiency is essential to effectively mitigate its negative impacts. Further in-depth studies examining the components of MetS and VD are also recommended.

## Figures and Tables

**Figure 1 nutrients-16-01480-f001:**
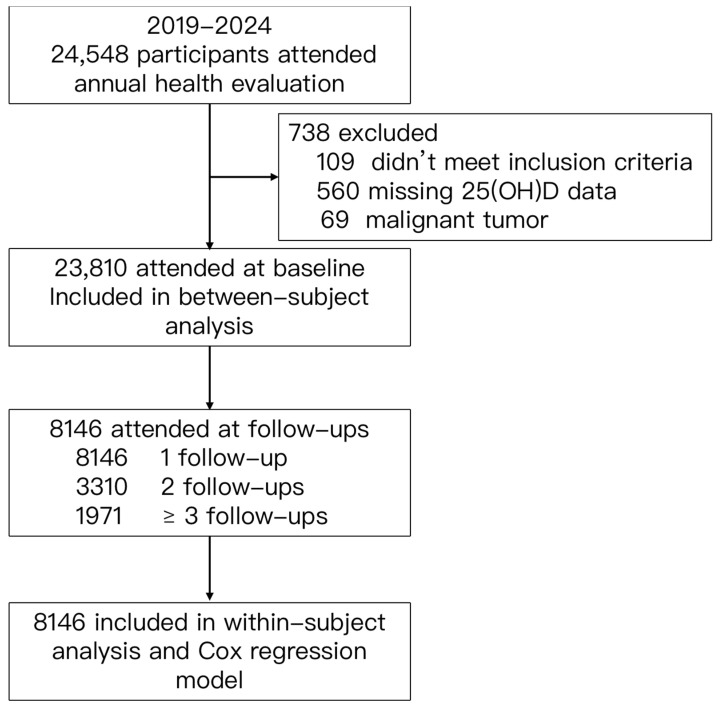
Flow chart of study participants.

**Figure 2 nutrients-16-01480-f002:**
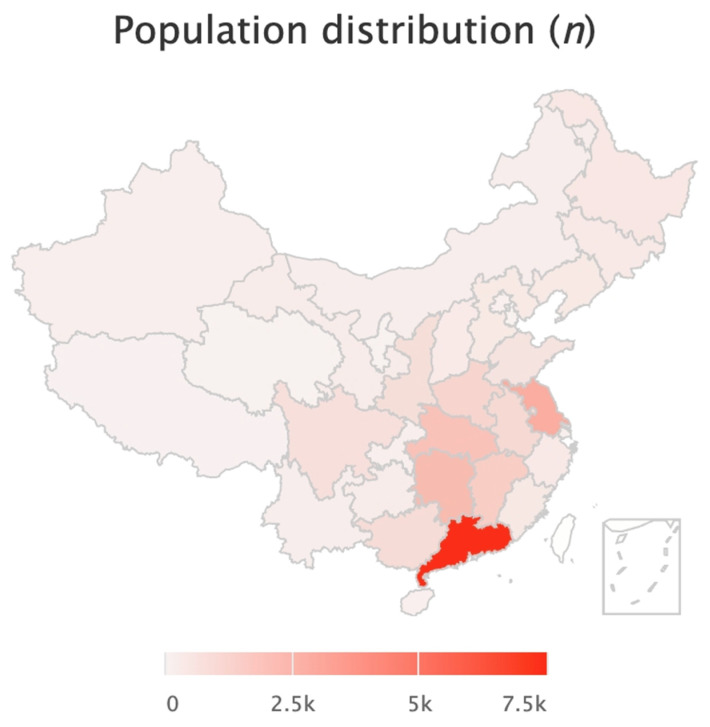
Geographical distribution of participants.

**Table 1 nutrients-16-01480-t001:** Baseline characteristics of study participants based on quartiles (Q) of serum 25(OH)D levels.

Variables	Overall	Q1	Q2	Q3	Q4	*p*-Value
*n*	23,810	6663	6172	5865	5110	
Sex, *n* (%)						0.490
Male	12,596 (52.9)	3539 (53.1)	3246 (52.6)	3068 (52.3)	2743 (53.7)	
Female	11,214 (47.1)	3124 (46.9)	2926 (47.4)	2797 (47.7)	2367 (46.3)	
Age, y	43.6 (13.3)	43.9 (13.3)	43.7 (13.4)	43.4 (13.3)	43.3 (13.2)	0.028
Age, *n* (%)						0.278
18–44	12,366 (51.9)	3384 (50.8)	3199 (51.8)	3093 (52.7)	2690 (52.6)	
45–59	8544 (35.9)	2442 (36.7)	2205 (35.7)	2069 (35.3)	1828 (35.8)	
≥60	2900 (12.2)	837 (12.6)	768 (12.4)	703 (12.0)	592 (11.6)	
Season, *n* (%)						<0.001
Spring	3879 (16.3)	1348 (20.2)	1019 (16.5)	862 (14.7)	650 (12.7)	
Summer	6131 (25.7)	1127 (16.9)	1510 (24.5)	1786 (30.5)	1708 (33.4)	
Fall	6139 (25.8)	1308 (19.6)	1526 (24.7)	1629 (27.8)	1676 (32.8)	
Winter	7661 (32.2)	2880 (43.2)	2117 (34.3)	1588 (27.1)	1076 (21.1)	
Smoking, *n* (%)	4852 (20.9)	1366 (21.2)	1195 (19.9)	1164 (20.3)	1127 (22.5)	0.005
Drinking, *n* (%)	6969 (30.1)	1948 (30.3)	1788 (29.8)	1675 (29.2)	1558 (31.1)	0.186
Systolic blood pressure, mmHg	122.3 (18.4)	122.7 (18.6)	122.3 (18.4)	122.2 (18.4)	122.1 (18.0)	0.335
Diastolic blood pressure, mmHg	71.8 (12.0)	72.2 (12.1)	71.8 (12.0)	71.7 (12.0)	71.5 (11.8)	0.008
Body mass index, kg/m^2^	23.8 (3.4)	23.9 (3.6)	23.9 (3.4)	23.8 (3.3)	23.6 (3.2)	<0.001
Waist circumference, cm	83.0 (10.2)	83.4 (10.8)	83.2 (10.3)	82.9 (10.0)	82.5 (9.5)	<0.001
25(OH)D, ng/mL	18.8 (7.3)	11.9 (3.1)	16.8 (3.6)	20.8 (4.2)	28.0 (6.6)	<0.001
Fasting glucose, mmol/L	5.6 (1.1)	5.7 (1.2)	5.6 (1.1)	5.6 (1.2)	5.6 (1.0)	<0.001
FBI, μU/mL	9.0 (14.3)	9.8 (21.7)	9.0 (13.1)	8.7 (9.6)	8.2 (5.3)	0.001
HOMA-IR	2.4 (5.0)	2.7 (7.4)	2.5 (5.1)	2.3 (3.0)	2.2 (1.7)	0.002
HbA1c, %	5.7 (0.9)	5.7 (0.9)	5.7 (0.9)	5.7 (0.9)	5.6 (0.8)	<0.001
Total cholesterol, mmol/L	5.0 (1.0)	5.0 (1.0)	5.0 (1.0)	5.0 (0.9)	5.0 (1.0)	0.006
LDL-C, mmol/L	3.1 (0.9)	3.0 (0.9)	3.1 (0.9)	3.1 (0.8)	3.1 (0.9)	0.028
HDL-C, mmol/L	1.4 (0.3)	1.3 (0.3)	1.4 (0.3)	1.4 (0.3)	1.4 (0.3)	<0.001
Triglycerides, mmol/L	1.5 (1.4)	1.6 (1.8)	1.5 (1.3)	1.5 (1.2)	1.4 (1.0)	<0.001
Metabolic syndrome, *n* (%)	6231 (27.5)	1907 (30.1)	1680 (28.7)	1485 (26.6)	1159 (23.8)	<0.001
Abdominal obesity, *n* (%)	9307 (40.3)	2651 (41.4)	2501 (41.9)	2248 (39.2)	1907 (38.1)	<0.001
High Triglycerides, *n* (%)	6260 (26.7)	1853 (28.6)	1717 (28.3)	1506 (26.0)	1184 (23.3)	<0.001
Low HDL-C, *n* (%)	4550 (21.9)	1383 (23.5)	1220 (22.6)	1101 (21.7)	846 (19.4)	<0.001
Hypertension, *n* (%)	8055 (34.4)	2284 (35.2)	2079 (34.3)	1988 (34.3)	1704 (33.7)	0.340
Hyperglycemia, *n* (%)	8829 (37.7)	2588 (39.9)	2284 (37.6)	2158 (37.2)	1799 (35.4)	<0.001

Data are shown as mean (SD) or number (%). *p*-values were derived from one-way ANOVA or Pearson’s Chi-squared test. Q1–Q4:. Age-and sex-specific quartiles of serum 25 hydroxyvitamin D.

**Table 2 nutrients-16-01480-t002:** Odds ratio of MetS and its components according to serum 25(OH)D quartiles *.

Variables	Between-Subject		Within-Subject	
	Model 1 OR (95% CI)	Model 2 OR (95% CI)	Model 1 OR (95% CI)	Model 2 OR (95% CI)
Metabolic syndrome				
Q1	1.00 (reference)	1.00 (reference)	1.00 (reference)	1.00 (reference)
Q2	0.77 (0.63, 0.94)	0.78 (0.65, 0.95)	0.21 (0.17, 0.25)	0.76 (0.63, 0.90)
Q3	0.58 (0.48, 0.72)	0.61 (0.51, 0.75)	0.30 (0.24, 0.36)	0.63 (0.52, 0.75)
Q4	0.38 (0.31, 0.47)	0.43 (0.35, 0.52)	0.28 (0.23, 0.34)	0.60 (0.50, 0.73)
*p*-trend	<0.001	<0.001	<0.001	<0.001
Abdominal obesity				
Q1	1.00 (reference)	1.00 (reference)	1.00 (reference)	1.00 (reference)
Q2	0.84 (0.63, 1.11)	0.85 (0.64, 1.13)	0.13 (0.10, 0.17)	0.34 (0.26, 0.46)
Q3	0.54 (0.40, 0.71)	0.54 (0.40, 0.72)	0.16 (0.12, 0.22)	0.29 (0.21, 0.39)
Q4	0.37 (0.28, 0.49)	0.40 (0.29, 0.54)	0.15 (0.11, 0.20)	0.26 (0.19, 0.36)
*p*-trend	<0.001	<0.001	<0.001	<0.001
HTG				
Q1	1.00 (reference)	1.00 (reference)	1.00 (reference)	1.00 (reference)
Q2	0.99 (0.84, 1.18)	0.99 (0.84, 1.16)	0.46 (0.40, 0.54)	0.90 (0.77, 1.05)
Q3	0.70 (0.59, 0.84)	0.71 (0.60, 0.84)	0.54 (0.46, 0.64)	0.79 (0.67, 0.93)
Q4	0.48 (0.40, 0.57)	0.49 (0.41, 0.58)	0.53 (0.44, 0.62)	0.78 (0.66, 0.93)
*p*-trend	<0.001	<0.001	<0.001	<0.001
Low HDL-C				
Q1	1.00 (reference)	1.00 (reference)	1.00 (reference)	1.00 (reference)
Q2	0.82 (0.67, 1.00)	0.83 (0.68, 1.02)	0.89 (0.74, 1.06)	1.08 (0.89, 1.30)
Q3	0.62 (0.51, 0.76)	0.63 (0.51, 0.77)	0.82 (0.67, 0.99)	0.92 (0.75, 1.12)
Q4	0.47 (0.38, 0.57)	0.48 (0.39, 0.59)	0.78 (0.64, 0.95)	0.87 (0.71, 1.07)
*p*-trend	<0.001	<0.001	<0.001	0.002
Hypertension				
Q1	1.00 (reference)	1.00 (reference)	1.00 (reference)	1.00 (reference)
Q2	0.88 (0.71, 1.09)	0.89 (0.74, 1.08)	0.08 (0.07, 0.10)	0.53 (0.44, 0.63)
Q3	0.86 (0.69, 1.07)	0.92 (0.76, 1.12)	0.20 (0.16, 0.25)	0.60 (0.50, 0.72)
Q4	0.81 (0.64, 1.01)	0.92 (0.76, 1.12)	0.16 (0.13, 0.20)	0.49 (0.41, 0.59)
*p*-trend	0.001	0.147	<0.001	<0.001
Hyperglycemia				
Q1	1.00 (reference)	1.00 (reference)	1.00 (reference)	1.00 (reference)
Q2	0.82 (0.70, 0.97)	0.81 (0.69, 0.94)	0.27 (0.23, 0.31)	0.88 (0.76, 1.01)
Q3	0.78 (0.66, 0.93)	0.80 (0.68, 0.93)	0.41 (0.35, 0.48)	0.80 (0.69, 0.93)
Q4	0.65 (0.54, 0.77)	0.70 (0.59, 0.82)	0.37 (0.31, 0.43)	0.75 (0.64, 0.87)
*p*-trend	<0.001	<0.001	<0.001	<0.001

*: Odds ratios (ORs) with 95% confidence intervals (CIs) for between-subjects and within-subject comparisons represent the between-subject (cross-sectional variations) and within-subject (longitudinal changes) components of the relationship calculated using hybrid logistic mixed models. The comparisons were made for age- and sex-specific quartiles (Q) 2 through 4 relative to quartile 1 (Q1) of 25-Hydroxyvitamin D levels. *p*-trend values were calculated by treating the quartiles as a continuous variable. Model 1: crude model; Model 2: adjustment for age, sex, collection season (spring, summer, fall, and winter), smoking (yes/no), and alcohol consumption (yes/no).

**Table 3 nutrients-16-01480-t003:** Hazard ratios (HRs) and 95% confidence intervals (CIs) for MetS and its components by quartiles of average 25-hydroxyvitamin D levels across multiple time points.

Variables	N		Model 1		Model 2	
	Total	New Cases	HR (95% CI) *	*p*-Trend	HR (95% CI) *	*p*-Trend
Metabolic syndrome	8146	761		0.014		0.007
Q1	1719	140	1.00		1.00	
Q2	2125	213	1.15 (0.93, 1.42)		1.15 (0.93, 1.42)	
Q3	2157	218	1.07 (0.87, 1.32)		1.09 (0.88, 1.34)	
Q4	2145	190	0.80 (0.64, 0.99)		0.80 (0.65, 1.00)	
Abdominal obesity	8146	786		0.003		0.002
Q1	1719	148	1.00		1.00	
Q2	2125	206	1.11 (0.90, 1.38)		1.11 (0.90, 1.37)	
Q3	2157	225	0.98 (0.79, 1.20)		0.98 (0.80, 1.21)	
Q4	2145	207	0.77 (0.63, 0.96)		0.77 (0.63, 0.96)	
High triglycerides	8146	872		0.013		0.011
Q1	1719	177	1.00		1.00	
Q2	2125	211	0.94 (0.77, 1.15)		0.94 (0.77, 1.14)	
Q3	2157	263	1.04 (0.86, 1.25)		1.02 (0.85, 1.24)	
Q4	2145	221	0.76 (0.63, 0.93)		0.76 (0.62, 0.92)	
Low HDL-C	8146	575		<0.001		<0.001
Q1	1719	137	1.00		1.00	
Q2	2125	143	0.80 (0.63, 1.01)		0.80 (0.64, 1.02)	
Q3	2157	150	0.75 (0.60, 0.95)		0.76 (0.61, 0.96)	
Q4	2145	145	0.63 (0.50, 0.80)		0.64 (0.50, 0.81)	
Hypertension	8146	824		0.323		0.328
Q1	1719	154	1.00		1.00	
Q2	2125	217	1.07 (0.87, 1.32)		1.06 (0.86, 1.31)	
Q3	2157	239	1.15 (0.94, 1.40)		1.16 (0.95, 1.43)	
Q4	2145	214	0.89 (0.72, 1.09)		0.89 (0.73, 1.10)	
Hyperglycemia	8146	1027		0.753		0.733
Q1	1719	204	1.00		1.00	
Q2	2125	254	1.01 (0.84, 1.21)		1.00 (0.83, 1.21)	
Q3	2157	284	1.01 (0.85, 1.21)		1.02 (0.85, 1.23)	
Q4	2145	285	0.96 (0.80, 1.15)		0.97 (0.81, 1.16)	

*: Cox regression analysis was used to calculate hazard ratios (HRs) and 95% confidence intervals (CIs) of metabolic syndrome incidence in quartiles 2–4 (vs. 1) of 25(OH)D. Quartile values were used as a continuous variable to calculate *p*-trend values. Q1–Q4: Age- and sex-specific quartiles 1–4 of average 25(OH)D levels across multiple time points before MetS onset among MetS-free individuals at baseline were used in the Cox regression model. Model 1: adjusted for sex and age. Model 2: adjusted for sex, age, season, smoking, and drinking.

## Data Availability

The original contributions presented in the study are included in the article/Appendix A, further inquiries can be directed to the corresponding authors.

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
