# Peer review of "Relationship between Circulating 25-Hydroxyvitamin D and Metabolic Syndrome in Chinese Adults: A Large Nationwide Longitudinal Study"

_nutrients, 2024, doi:10.3390/nu16101480_

Round 1

Reviewer 1 Report

Comments and Suggestions for Authors

The sample size of this work is really important; The large number of the enrolled population should allow clinically important conclusions to be reached.

However, I have some suggestions for the authors:

1. They should revise the Introduction section, focusing their attention on the possible pathogenetic mechanisms that correlate the components of metabolic syndrome to hypovitaminosis D. I think it's important to refer to the mild inflammation that characterizes patients with low vitamin D levels.

2. Not surprisingly, hypovitaminosis D is not associated with hyperglycemia; In fact, it is preceded by many years of insulin resistance which, in my opinion, is the mechanism that unifies all the components of the metabolic syndrome. In this regard, I suggest that the following works be listed: Cardiovasc Diabetol. 2014 Feb 20;13:48. doi: 10.1186/1475-2840-13-48.PMID: 24555478 and  Molecules. 2019 Jul 9;24(13):2499. doi: 10.3390/molecules24132499. PMID: 31323907. 

3.The same considerations apply to the blood pressure values.

4. I note that many metabolic variables, as well as the BP values, do not show a biologically plausible difference in vitamin D quintiles; They need to explain this phenomenon and revise the whole discussion based on these considerations. The large population makes it possible to achieve statistical significance that is not necessarily associated with biological plausibility.

Comments on the Quality of English Language

An extensive revision is needed

Author Response

We appreciate the opportunity to revise our manuscript entitled " Relationship between Serum 25-Hydroxyvitamin D and Metabolic Syndrome in Chinese Adults: A Comprehensive Nationwide Longitudinal Study” (Manuscript ID: 2971049)" and thank the reviewers for their insightful comments and suggestions. We have carefully considered each comment and made corresponding revisions to the manuscript. Below, we detail our responses to the comments and the changes made to the manuscript.

Should there be any queries regarding this paper, please feel free to contact me.

Sincerely yours,                                                                 

Yu-Ming Chen for the authors

Professor

Department of Epidemiology

School of Public Health

Sun Yat-sen University

Point-to-point response to the Reviews comments

Comment 1: They should revise the Introduction section, focusing their attention on the possible pathogenetic mechanisms that correlate the components of metabolic syndrome to hypovitaminosis D. I think it's important to refer to the mild inflammation that characterizes patients with low vitamin D levels.

Response 1: Thank you for your insightful comment regarding the need to elaborate on the pathogenetic mechanisms connecting hypovitaminosis D with metabolic syndrome. We have revised the Introduction to include a detailed discussion of the potential pathogenic mechanisms and the mild inflammation commonly observed in patients with low vitamin D levels. This update, outlined in lines 44-59, is supported by additional references that enhance the section's comprehensiveness.

Changes in the Main text:“The effect of vitamin D on MetS may be related to the following factors. 1) In certain instances, a deficiency in vitamin D is linked to endothelial dysfunction[6]. VD enhances nitric oxide levels, facilitating blood vessel dilation. VD deficiency can impact endothelial cells, leading to vascular dysfunction and disrupting vascular homeostasis[7]. 2) Reducing insulin resistance attributed to VD may involve phosphorylation of IRS1 at Ser307, enhanced expression of Peroxisome proliferator-activated receptor gamma (PPAR-γ), and decreased phosphorylation of nuclear factor kappa B (NF-κB) p65 at Ser536[8]. 3)Interaction between vitamin D and the VD receptor (VDR) reduces the levels of the proinflammatory cytokines, interleukin (IL)-2 and IL-12[9]. 4) VD exerts a significant impact on lipid profiles through the suppression of 3-Hydroxy-3-methylglutaryl-coenzyme A (HMG-CoA) reductase (HMGCR) enzymatic activity, which may enhance lipid metabolism in MetS [10, 11]. Additionally, VD can serve as a fatty acid binding protein 4 (FABP4) inhibitor, offering therapeutic benefits for MetS by reducing bile secretion and regulating the PPAR signaling pathway[12].”

Comment 2: Not surprisingly, hypovitaminosis D is not associated with hyperglycemia; In fact, it is preceded by many years of insulin resistance which, in my opinion, is the mechanism that unifies all the components of the metabolic syndrome. In this regard, I suggest that the following works be listed: Cardiovasc Diabetol. 2014 Feb 20;13:48. doi: 10.1186/1475-2840-13-48.PMID: 24555478 and  Molecules. 2019 Jul 9;24(13):2499. doi: 10.3390/molecules24132499. PMID: 31323907. 
The same considerations apply to the blood pressure values.

Response 2: We greatly value your suggestions and have incorporated the recommended citations regarding the relationship between vitamin D and hyperglycemia into the revised manuscript, specifically in the discussion section (lines 330-346). These additions deepen our analysis of how hypovitaminosis D precedes insulin resistance and influences hyperglycemia, along with the associated blood pressure implications. 

Changes in the Main text:

“4.4. Serum 25(OH)D and Hyperglycemia

In the investigation of the association between VD and hyperglycemia, it was observed that FBG, FBI, HOMA-IR, HbA1c, and PBG levels were increased in participants with the highest levels (Q4 vs. Q1) of serum 25(OH)D. Both cross-sectional and longitudinal studies have documented a significant correlation between serum 25(OH)D and hyperglycemia. Cross-sectional studies in 3289 Chinese middle-aged individuals [26]and in 8,421 American men and nonpregnant women [45] revealed a significant relationship between low 25(OH)D concentrations and increased risk of insulin resistance. Cohort studies found lower concentrations of 25(OH)D were associated with increased fasting glucose and insulin resistance in 524 non-diabetic individuals aged 40 to 69 years in the UK [56] and 4,164 Australian adults [57]. Sciacqua et al. [58] also identified a negative correlation between 25(OH)D concentrations and glucose tolerance. VD influences glucose and insulin homeostasis by modulating β-cell function and immune responses, enhancing peripheral insulin sensitivity, and ameliorating pro-inflammatory conditions [59]. Related studies suggest elevated PTH [34, 58] and obesity [60] may also impact VD's biological effects. The biological effects of vitamin D are multifactorial and complex. ”

Comment 3: I note that many metabolic variables, as well as the BP values, do not show a biologically plausible difference in vitamin D quintiles; They need to explain this phenomenon and revise the whole discussion based on these considerations. The large population makes it possible to achieve statistical significance that is not necessarily associated with biological plausibility.
Response 3: Thank you for pointing out the issues regarding the lack of biologically plausible differences across vitamin D quintiles for certain metabolic variables and blood pressure values. We have addressed this by carefully revising the discussion to better explain these phenomena, considering the large sample size and its implications for statistical significance versus biological relevance. This revised analysis is now included in the hypertension section of our discussion (lines 348-360).

Changes in the Main text:

 “4.5. Serum 25(OH)D and Hypertension

Our longitudinal analysis found a fascinating negative correlation between serum 25(OH)D concentrations and the risk of hypertension. Consistent findings were observed regarding the beneficial relationship between vitamin D and hypertension [57, 61-63]. A study on the removal of the vitamin D receptor (VDR) gene in mice showed heightened activity in the renin-angiotensin-aldosterone system (RAAS) and elevated blood pressure levels [64]. However, some studies also found null associations. A meta-analysis of RCTs found that vitamin D supplementation did not affect blood pressure in children and adolescents [65]. A 10% increase in serum 25(OH)D levels was not associated with blood pressure (SBP) in 146,581 Europeans [66]. Therefore, large-sample, multicenter, prospective randomized controlled trials are warranted to understand further the complex relationship between vitamin D and blood pressure. ”

Reference

  1. Tarcin O, Yavuz DG, Ozben B, Telli A, Ogunc AV, Yuksel M, Toprak A, Yazici D, Sancak S, Deyneli O, Akalin S: Effect of vitamin D deficiency and replacement on endothelial function in asymptomatic subjects.J Clin Endocrinol Metab 2009, 94:4023-4030.
  2. Tran V, De Silva TM, Sobey CG, Lim K, Drummond GR, Vinh A, Jelinic M: The Vascular Consequences of Metabolic Syndrome: Rodent Models, Endothelial Dysfunction, and Current Therapies. Front Pharmacol 2020, 11:148.
  3. Krisnamurti DGB, Louisa M, Poerwaningsih EH, Tarigan TJE, Soetikno V, Wibowo H, Nugroho CMH: Vitamin D supplementation alleviates insulin resistance in prediabetic rats by modifying IRS-1 and PPARγ/NF-κB expressions.Front Endocrinol (Lausanne) 2023, 14:1089298.
  4. Surdu AM, Pînzariu O, Ciobanu D-M, Negru A-G, Căinap S-S, Lazea C, Iacob D, Săraci G, Tirinescu D, Borda IM, Cismaru G: Vitamin D and Its Role in the Lipid Metabolism and the Development of Atherosclerosis.Biomedicines 2021, 9:172.
  5. Faraji S, Alizadeh M: Mechanistic Effects of Vitamin D Supplementation on Metabolic Syndrome Components in Patients with or without Vitamin D Deficiency. J Obes Metab Syndr 2020, 29:270-280.
  6. Gupta AK, Sexton RC, Rudney H: Effect of vitamin D3 derivatives on cholesterol synthesis and HMG-CoA reductase activity in cultured cells. J Lipid Res 1989, 30:379-386.
  7. Xia Y, Yu Y, Zhao Y, Deng Z, Zhang L, Liang G: Insight into the Interaction Mechanism of Vitamin D against Metabolic Syndrome: A Meta-Analysis and In Silico Study. Foods 2023, 12:3973.
  8. Lu L, Yu Z, Pan A, Hu FB, Franco OH, Li H, Li X, Yang X, Chen Y, Lin X: Plasma 25-hydroxyvitamin D concentration and metabolic syndrome among middle-aged and elderly Chinese individuals. Diabetes Care 2009, 32:1278-1283.
  9. Reis JP, von Mühlen D, Kritz-Silverstein D, Wingard DL, Barrett-Connor E: Vitamin D, parathyroid hormone levels, and the prevalence of metabolic syndrome in community-dwelling older adults. Diabetes Care 2007, 30:1549-1555.
  10. Ford ES, Ajani UA, McGuire LC, Liu S: Concentrations of serum vitamin D and the metabolic syndrome among U.S. adults. Diabetes Care 2005, 28:1228-1230.
  11. Forouhi NG, Luan J, Cooper A, Boucher BJ, Wareham NJ: Baseline serum 25-hydroxy vitamin d is predictive of future glycemic status and insulin resistance: the Medical Research Council Ely Prospective Study 1990-2000. Diabetes 2008, 57:2619-2625.
  12. Gagnon C, Lu ZX, Magliano DJ, Dunstan DW, Shaw JE, Zimmet PZ, Sikaris K, Ebeling PR, Daly RM: Low serum 25-hydroxyvitamin D is associated with increased risk of the development of the metabolic syndrome at five years: results from a national, population-based prospective study (The Australian Diabetes, Obesity and Lifestyle Study: AusDiab). J Clin Endocrinol Metab 2012, 97:1953-1961.
  13. Sciacqua A, Perticone M, Grillo N, Falbo T, Bencardino G, Angotti E, Arturi F, Parlato G, Sesti G, Perticone F: Vitamin D and 1-hour post-load plasma glucose in hypertensive patients.Cardiovasc Diabetol 2014, 13:48.
  14. Kabadi SM, Lee BK, Liu L: Joint effects of obesity and vitamin D insufficiency on insulin resistance and type 2 diabetes: results from the NHANES 2001-2006. Diabetes Care 2012, 35:2048-2054.
  15. Vimaleswaran KS, Berry DJ, Lu C, Tikkanen E, Pilz S, Hiraki LT, Cooper JD, Dastani Z, Li R, Houston DK, et al: Causal relationship between obesity and vitamin D status: bi-directional Mendelian randomization analysis of multiple cohorts.PLoS Med 2013, 10:e1001383.
  16. Wang TJ, Pencina MJ, Booth SL, Jacques PF, Ingelsson E, Lanier K, Benjamin EJ, D'Agostino RB, Wolf M, Vasan RS: Vitamin D deficiency and risk of cardiovascular disease.Circulation 2008, 117:503-511.
  17. Zhao G, Ford ES, Li C, Kris-Etherton PM, Etherton TD, Balluz LS: Independent associations of serum concentrations of 25-hydroxyvitamin D and parathyroid hormone with blood pressure among US adults. J Hypertens 2010, 28:1821-1828.
  18. Qi KJ, Zhao ZT, Zhang W, Yang F: The impacts of vitamin D supplementation in adults with metabolic syndrome: A systematic review and meta-analysis of randomized controlled trials. Front Pharmacol 2022, 13:1033026.
  19. Jensen NS, Wehland M, Wise PM, Grimm D: Latest Knowledge on the Role of Vitamin D in Hypertension.International Journal of Molecular Sciences 2023, 24:4679.
  20. Cai B, Luo X, Zhang P, Luan Y, Cai X, He X: Effect of vitamin D supplementation on markers of cardiometabolic risk in children and adolescents: A meta-analysis of randomized clinical trials. Nutr Metab Cardiovasc Dis 2021, 31:2800-2814.
  21. Vimaleswaran KS, Cavadino A, Berry DJ, Jorde R, Dieffenbach AK, Lu C, Alves AC, Heerspink HJ, Tikkanen E, Eriksson J, et al: Association of vitamin D status with arterial blood pressure and hypertension risk: a mendelian randomisation study.Lancet Diabetes Endocrinol 2014, 2:719-729.

Reviewer 2 Report

Comments and Suggestions for Authors

The paper "Relationship between serum 25-hydroxyvitamin D and metabolic syndrome in Chinese adults: A large nationwide longitudinal study" addresses an important issue. Researchers presented an interesting study examining the association between serum 25(OH)D levels and the risk of metabolic syndrome and its components in a nationwide cohort involving 23,810 adults. However, upon review, several points require clarification and expansion. Please find my comments and suggestions below:

1. Methods: Although it is a study of interest and importance, the study lacks methodological quality in terms of controlling and identifying biases, as well as in the design and methodology. In the Research Method section (2.2.1 – Data Collection), please add criteria for including and excluding participants from the study. Additionally, please add a new Figure - a flow chart of the selection of study participants. I recommend including details of these criteria to provide comprehensive information in this section.

2. Results: Please supplement the information regarding vitamin D status, following the criteria outlined in section 2.3. Relevant Definitions (lines 143-147). Present the obtained results in the main part of the paper. The authors suggest including information on this topic in Table S1 (line 185). Unfortunately, the reviewer does not see the data mentioned by the authors in the text (lines 179-185). In the reviewer's opinion, information regarding the assessment of Vitamin D status is important and should be included in the main part of the paper.

3. Discussion: In this section, please add a new subsection "Prevalence of Vitamin D Deficiency". In this part, discuss the obtained research results in comparison to the results obtained by other authors. Including these insights would enrich the discussion and provide a better understanding of the research findings.

4. Discussion: Due to the crucial impact of vitamin D supplementation on vitamin D status, please also consider this element in the discussion and present what percentage of the Chinese adult population (18 years of age and older) uses vitamin D supplements and what is their effectiveness.

5. At the end of the Discussion section, please further emphasize the following themes: implications of this study and novelty in this study.

Comments on the Quality of English Language

Minor editing of English language is required.

Author Response

We appreciate the opportunity to revise our manuscript entitled " Relationship between Serum 25-Hydroxyvitamin D and Metabolic Syndrome in Chinese Adults: A Comprehensive Nationwide Longitudinal Study” (Manuscript ID: 2971049)" and thank the reviewers for their insightful comments and suggestions. We have carefully considered each comment and made corresponding revisions to the manuscript. Below, we detail our responses to the comments and the changes made to the manuscript.

Should there be any queries regarding this paper, please feel free to contact me.

Sincerely yours,                                                                 

Yu-Ming Chen for the authors

Professor

Department of Epidemiology

School of Public Health

Sun Yat-sen University

Point-to-point response to the Reviews comments

Comment 1: Methods: Although it is a study of interest and importance, the study lacks methodological quality in terms of controlling and identifying biases, as well as in the design and methodology. In the Research Method section (2.2.1 – Data Collection), please add criteria for including and excluding participants from the study. Additionally, please add a new Figure - a flow chart of the selection of study participants. I recommend including details of these criteria to provide comprehensive information in this section.

Response 1: Thank you for your valuable suggestions. We have revised the Research Method section to include detailed inclusion and exclusion criteria for study participants. These criteria ensure a comprehensive and representative sample, enhancing the study’s methodological rigor. We have also incorporated a flowchart illustrating the participant selection process, along with a map showing the geographical distribution of the participants, which provides clear visual aids to understand the scope of our study.

Changes in the Main text:

“2.1. Study Participants

This cohort study encompassed participants who underwent annual health evaluations at the Yibicom Health Management Center in Guangzhou, China, covering five years from 2019 to 2024. Eligible participants are also required to be Chinese residents born in China, with an age of 18 or above. A total of 24,548 participants attended the health evaluations from 30 different cities across the country. Among them, we excluded those with missing data of MetS status or serum 25(OH)D (n= 560), 109 cases who didn’t meet the requirement of residences or the age range, and 69 cases with malignant tumor. 23,810 participants were included in the cross-sectional investigation. The longitudinal analyses contained the individuals with 8146, 3310, and 1971 completed two, three, and over three times of annual health evaluations (Fig. 1 & Fig.2). ”

Comment 2:  Results: Please supplement the information regarding vitamin D status, following the criteria outlined in section 2.3. Relevant Definitions (lines 143-147). Present the obtained results in the main part of the paper. The authors suggest including information on this topic in Table S1 (line 185). Unfortunately, the reviewer does not see the data mentioned by the authors in the text (lines 179-185). In the reviewer's opinion, information regarding the assessment of Vitamin D status is important and should be included in the main part of the paper.

Response 2: We apologize for any confusion caused by the previous layout of our manuscript. Based on your feedback, we have included detailed information on vitamin D status in the Results section and have updated Table 1 to summarize participant characteristics by their vitamin D levels. This change ensures that key data are accessible and highlighted within the main body of the text.

Changes in the Main text:  “

3.1. Participant Characteristics

The investigation involved 23,810 participants, comprising 12,596 males and 11,214 females. The mean (SD) values of age and serum 25(OH)D were 43.6 (13.3) years and 18.8 (7.3) ng/mL. 17.9% (n=4260), 42.9% (n=10213), and 39.2% (n=9337) of participants were  VD deficiency, insufficiency, and sufficiency based on serum 25(OH)D, respectively. VD deficiency and insufficiency were more frequent in women than men (64.0% vs. 57.9%).”

Comment 3: Discussion: In this section, please add a new subsection "Prevalence of Vitamin D Deficiency". In this part, discuss the obtained research results in comparison to the results obtained by other authors. Including these insights would enrich the discussion and provide a better understanding of the research findings.

Response 3: We value your recommendation to expand the discussion on vitamin D deficiency. We have introduced a new subsection titled “Prevalence of Vitamin D Deficiency and Insufficiency and Vitamin D Supplement Users in Chinese Adults.” This section contextualizes our findings within the broader spectrum of existing research and discusses variances in vitamin D status among different demographic groups in China.

Changes in the Main text:  “

4.1. Prevalence of VD deficiency and insufficiency and VD supplement users in Chinese adults

Many studies report a prevalent VD deficiency and insufficiency in Chinese adults. Our findings showed a 17.9% VD deficiency and 42.9% VD insufficiency among the middle-aged population. A notable variance in the prevalence of VD deficiency and insufficiency among various Chinese populations, such as 28.6% deficiency and 37.6% insufficiency in urban middle-aged individuals aged 35-60 by Yin et al. [39], 78.3% deficiency (< 20 ng/ml) in adolescents aged 14-28[30], and 69.2% and 24.4% of deficiency and insufficiency among the elderly population aged 50-70 [26], respectively. This discrepancy may be due to the inconsistency in age and place of residence of the research participants. …”

Comment 4: Discussion: Due to the crucial impact of vitamin D supplementation on vitamin D status, please also consider this element in the discussion and present what percentage of the Chinese adult population (18 years of age and older) uses vitamin D supplements and what is their effectiveness. 

Response 4: Thank you for highlighting the importance of vitamin D supplementation in the discussion. We acknowledge the limited availability of data on vitamin D supplement use among the general adult population in China, which has been focused more on specific groups like pregnant women and children. We have noted this gap in the literature and outlined the need for future research to address this area.

Changes in the Main text:  “

4.1. Prevalence of VD deficiency and insufficiency and VD supplement users in Chinese adults

VD insufficiency is prevalent in the Chinese population. However, domestic surveys on VD supplementation have focused on children[40], pregnant women [41, 42],with fewer surveys on young or middle-aged adults, with a proportion of the VD supplement usage ranged between 4.8% to 50.9%. Our study did not collect data on supplement use, thereby limiting our ability to assess the impact of vitamin D supplements on serum levels and the prevalence of deficiency and insufficiency. Considering substantially high VD deficiency and insufficiency in this population, the current VD usage isn’t enough to control this problem. Future studies should aim to fill this gap by examining the effects of supplementation across a broader demographic.”

Comment 5: At the end of the Discussion section, please further emphasize the following themes: implications of this study and novelty in this study.

Response 5: We appreciate your recommendation to enhance the emphasis on the study's implications and its novelty. We have revised the Conclusion section to highlight these aspects more distinctly, ensuring that the unique contributions and future directions proposed by our study are clearly articulated.

Changes in the Main text:  “

  1. Conclusion

Our study elucidated that lower serum 25(OH)D concentrations are linked to raised risks of MetS and several of its components in young and middle-aged adults in China. The robust evidence provided by our longitudinal study design, which utilized a large nationwide sample, supports these findings. However, further well-designed randomized controlled trials are necessary to establish a causal relationship. Additionally, greater focus on the adverse effects of vitamin D deficiency is essential to effectively mitigate its negative impacts. Further in-depth studies examining the components of MetS and vitamin D are also recommended.”

Reference

  1. Lu L, Yu Z, Pan A, Hu FB, Franco OH, Li H, Li X, Yang X, Chen Y, Lin X: Plasma 25-hydroxyvitamin D concentration and metabolic syndrome among middle-aged and elderly Chinese individuals. Diabetes Care 2009, 32:1278-1283.
  2. Fu J, Han L, Zhao Y, Li G, Zhu Y, Li Y, Li M, Gao S, Willi SM: Vitamin D levels are associated with metabolic syndrome in adolescents and young adults: The BCAMS study. Clin Nutr 2019, 38:2161-2167.
  3. Yin X, Sun Q, Zhang X, Lu Y, Sun C, Cui Y, Wang S: Serum 25(OH)D is inversely associated with metabolic syndrome risk profile among urban middle-aged Chinese population. Nutr J 2012, 11:68.
  4. Ke N, Jin-Zi W, Huan W: Investigation on the influencing factors of calcium and vitamin D supplementations in  children from 9 areas of China. Maternal and Child Health Care of China 2014,32:5289-5293.
  5. Yin WJ, Tao RX, Hu HL, Zhang Y, Jiang XM, Zhang MX, Jin D, Yao MN, Tao FB, Zhu P: The association of vitamin D status and supplementation during pregnancy with gestational diabetes mellitus: a Chinese prospective birth cohort study. Am J Clin Nutr 2020, 111:122-130.
  6. Hu Y, Wang R, Mao D, Chen J, Li M, Li W, Yang Y, Zhao L, Zhang J, Piao J, et al: Vitamin D Nutritional Status of Chinese Pregnant Women, Comparing the Chinese National Nutrition Surveillance (CNHS) 2015-2017 with CNHS 2010-2012. Nutrients 2021, 13:2237.

Round 2

Reviewer 1 Report

Comments and Suggestions for Authors

First of all, I would like to thank the authors for the thorough revision carried out in the various sections of the manuscript; however, I still have a few comments.

1. Table 1 can be eliminated because it adds nothing to Table 2 where the different parameters are reported according to vitamin D quartiles.

2.In the statistical analysis section, reference is made only to OR and not to HR; please check this section and explain the different meaning of the two parameters.

3. Please give a biologically plausible explanation of the results in Table 4. In fact, for some parameters, HR has a rollercoaster trend; the significance of the P, therefore, is irrelevant.

Comments on the Quality of English Language

minor editing revision is required

Author Response

We feel great thanks for your professional review work on our article. According to your nice suggestions, We have submitted an updated version of our manuscript, revised in accordance with your recommendations, entitled “Relationship between Serum 25-Hydroxyvitamin D and Metabolic Syndrome in Chinese Adults: A Comprehensive Nationwide Longitudinal Study” (Manuscript ID: 2971049). The modifications made to the manuscript are evident in this revised version.

Should you have any queries regarding this paper, please feel free to contact me.

Sincerely yours,                                                                 

Yu-Ming Chen for the authors

Professor

Department of Epidemiology

School of Public Health

Sun Yat-sen University

Here are point-to-point answers to your detailed comments.

Comment 1: Table 1 can be eliminated because it adds nothing to Table 2 where the different parameters are reported according to vitamin D quartiles.

Response 1: Thank you for pointing out the duplication of data in Table 1 and Table 2. We moved Table 1 from the main body of the manuscript to Supplemental Material (Table S1)(lines 539).

Comment 2: In the statistical analysis section, reference is made only to OR and not to HR; please check this section and explain the different meaning of the two parameters.

Response 2: We apologize for not clarifying the meaning of OR and HR values in our previous manuscript. Based on your feedback, in the Statistical Analysis section and the footnotes of Tables 2 and 3, we have added different methods to explore the relationship between VD and metabolic syndrome (see lines 171-182). These additions have improved the rigor of our research methodology. Please refer to the changes in the Main text and table footnotes for the details.

Changes in the Main text:

“Survival analysis to explore the risk of MetS in quartiles 2-4 (vs. 1) of average 25(OH)D values before MetS onset in participants without MetS at baseline, and calculated hazard ratios (HRs) and 95% confidence intervals (CIs) by employing Cox regression in R studio (R Studio, Boston, MA, USA). A logistic hybrid mix-effect model was employed to capture both the intra- and inter-subject relationships between VD and MetS [38]. Odds ratios (ORs) and 95% confidence intervals (CIs) were calculated to describe the estimated effect sizes for the associations between the quartiles of intra-subject (i ) and inter-subject variations (i ) of 25(OH)D with the status of MetS (MetSt) at various time points (t) using “xtreg” program of Stata/SE 17.0 software (StataCorp LLC, TX, USA) which indicate cross-sectional and longitudinal relationships between VD and the presence of MetS (Equation 1) [38]. P-trend values were calculated by treating the quartiles as a continuous variable.”

Changes in the Footnotes

Table 2. “*: Odds Ratios (ORs) with 95% Confidence Intervals (CIs) for between-subjects and within-subject comparisons represent the between-subject (cross-sectional variations) and within-subject (longitudinal changes) components of the relationship calculated using hybrid logistic mixed models. The comparisons were made for age- and sex-specific quartiles (Q) 2 through 4 relative to quartile 1 (Q1) of 25-Hydroxyvitamin D levels. P-trend values were calculated by treating the quartiles as a continuous variable.”

Table 3. “*: Cox regression analysis was used to calculate hazard ratios (HRs) and 95% confidence intervals (CIs) of metabolic syndrome incidence in quartiles 2-4 (vs. 1) of 25(OH)D. Quartile values were used as a continuous variable to calculate P-trend values. Q1-Q4: Age- and sex- specific quartiles 1-4 of average 25(OH)D levels across multiple time points before MetS onset in participants without MetS at baseline were used in the Cox regression model.”

Comment 3: Please give a biologically plausible explanation of the results in Table 4. In fact, for some parameters, HR has a rollercoaster trend; the significance of the P, therefore, is irrelevant.

Response 3: Thank you for highlighting the absence of biologically plausible differences in certain metabolic variables across VD quartiles in the survival analysis presented in Table 3. Our findings using the Cox regression model indicated no significant reduction in HRs for MetS and some of its components in Quartiles 2 and 3 of VD, with some HRs even showing a slight increase  compared to Quartile 1 (P>0.05). A significant decrease in HRs was observed only in Quartile 4. In contrast, more pronounced beneficial associations between VD and MetS were detected using hybrid logistic mixed-effect models.

These discrepancies are thoroughly discussed in the discussion section. Notably, the beneficial association between VD and MetS is more marked in the ORs derived from hybrid logistic mixed models than in the HRs from the Cox regression models. The ORs were calculated using hybrid logistic mixed-effect models that consider variations in VD and MetS status throughout all repeated measurements. On the other hand, the Cox regression model identified incident cases based on the detection of MetS at any follow-up among participants initially free from MetS, relying on average VD levels prior to the onset of MetS. This method did not account for fluctuations in VD levels or subsequent changes in MetS status. Similar patterns were observed for certain MetS components, such as hypertension and hyperglycemia. Thus, hybrid logistic mixed-effect models likely provide a more robust framework for understanding the relationship between VD and MetS compared to Cox regression models.

Changes in the Main text:

4.2. Serum 25(OH)D and metabolic syndrome

“… Intriguingly, our findings indicate that the beneficial association between VD and MetS is more pronounced in the ORs derived from hybrid logistic mixed models than in the HRs from Cox regression models. The ORs were calculated using hybrid logistic mixed-effect models to account for changes in VD and MetS status across all repeated measurements. Conversely, the Cox regression model defined incident cases based on the detection of MetS at any follow-up visit among initially MetS-free individuals, using average VD levels prior to the onset of MetS. This approach overlooked fluctuations in VD levels and subsequent changes in MetS status. Similar situations were also noted for some MeS components (e.g., hypertension and hyperglycemia). Therefore, the hybrid logistic mixed-effect models likely offer more robust evidence of the VD-MetS relationship than the Cox regression models.”

4.3. Serum 25(OH)D and abdominal obesity and dyslipidemia

“...The influence of vitamin D on lipid profiles involves a multifaceted mechanism[56], encompassing both the activation of its receptors and the reduction of lipid accumulation. When serum vitamin D concentrations surpass a specific limit, its beneficial impacts on lipid parameters become significantly more evident.”

Reference

  1. Wang J, Liu Y: Research progress of mechanism of vitamin D in regulating obesity and insulin resistance on children. Chinese Journal of Applied Clinical Pediatrics 2015:1598-1600.

Reviewer 2 Report

Comments and Suggestions for Authors

The authors have addressed the questions raised in the first round of review. They have made adjustments to the manuscript based on the provided comments. As a results, I have no further comments on the revised manuscript.

Author Response

We feel great thanks for your valuable comments that made the revised manuscript more readable and scientifically sound.

Sincerely yours,                                                                 
Yu-Ming Chen for the authors
Professor
Department of Epidemiology
School of Public Health
Sun Yat-sen University